# Development of Inherent Vulnerability Index within Jammu Municipal Limits, India

Simran Bharti [1], Adyan Ul Haq [1], L. T. Sasang Guite [1], Shruti Kanga [1,*], Fayma Mushtaq [2], Majid Farooq [3], Suraj Kumar Singh [4], Pankaj Kumar [5,*] and Gowhar Meraj [6]

[1] Department of Geography, School of Environment and Earth Sciences, Central University, Bathinda 151401, India; simranbhagat400@gmail.com (S.B.); adyangeo@gmail.com (A.U.H.); ltsguite@cup.edu.in (L.T.S.G.)

[2] Applied Research Centre for Environmental and Marine Studies, King Fahd University of Petroleum and Minerals, Dhahran 31261, Saudi Arabia; fayma.mushtaq@kfupm.edu.sa

[3] Department of Ecology, Environment and Remote Sensing, Government of Jammu and Kashmir, Srinagar 190018, India; majid.farooq@jk.gov.in

[4] Centre for Climate Change and Water Research, Suresh Gyan Vihar University, Jaipur 302017, India; suraj.kumar@mygyanvihar.com

[5] Institute for Global Environmental Strategies, Hayama 240-0115, Japan

[6] Graduate School of Agricultural and Life Sciences, The University of Tokyo, 1-1-1 Yayoi, Tokyo 113-8654, Japan; gowharmeraj@g.ecc.u-tokyo.ac.jp

\* Correspondence: shruti.kanga@cup.edu.in (S.K.); kumar@iges.or.jp (P.K.)

**Abstract:** Evaluating inherent vulnerability, an intrinsic characteristic becomes imperative for the formulation of adaptation strategies, particularly in highly complex and vulnerable regions of Himalayas. Jammu City, situated in the north-western Himalayas within a transitional zone between the Himalayan range and the plains, is not only susceptible to intense seismic activities but also faces multiple hazards, including floods, earthquakes, avalanches, and landslides. In recent years, the region has experienced growth in population with rapid progress in infrastructure development, encompassing the construction of highways, dams, and tunnels as integral components of urban development initiatives. Therefore, this study has been conducted to assess the inherent vulnerability index (VI) in Jammu City at ward level as a function of sensitivity, adaptive capacity, and exposure, using ecological and social indicators in GIS environment. The primary objective was to identify the most vulnerable area and ascertain the corresponding municipal ward, aiming to formulate a comprehensive ranking. The 22 indicators analysed were from four major components, namely social, infrastructure, technological, and ecological. The ecological indicators like Land Surface Temperature (LST), Normalized Difference Vegetation Index (NDVI), and Land use/Land cover were derived from Landsat 8 OLI satellite data. The results show that the majority of the area of the city falls into the moderate (20%), high (25.49%), and very high (25.17%) vulnerability categories, respectively, clustered in north-western and south-western transects with densely populated residential areas. The results can assist policymakers in identification of components of inherent vulnerability for focused resource management and formulating adaptation strategies to address the current stressors in the region.

**Keywords:** indicator-based method; inherent vulnerability; IPCC; Jammu City; NDVI; ward level; resilience

## 1. Introduction

According to the United Nations in 2019, the world's population keeps on growing steadily and is anticipated to reach 9.7 billion by 2050 [1]. Out of this, 68% of the global population is projected to live in cities, with almost 90% of urban expansion occurring in the least developed regions of Asia and Africa. The physical expansion of urban areas is surpassing population growth, with urban areas increasing at even faster rates than

their corresponding populations [2,3]. India, a lower-middle-income country, is anticipated to witness a substantial rise in its urban population and its challenges for sustainable development, with projections suggesting an addition of 404 million urban residents by 2050, almost doubling the current numbers [4]. The continuous increase in global urban population, driven by economic growth and industrialization, is closely linked to most environmental issues worldwide and, coupled with competition for resource access, is likely to impact population vulnerability [5]. In addition, urbanization, a non-climate factor, is accompanied by the detrimental effects of climate change, impacting water and food availability, agricultural produce, and exacerbating poverty [6]. This close link contributes to a synergistic 'negative' impact and pushes the global community toward higher exposure to natural hazards) [7–10]. Globally, approximately 500,000 residents are reported to be substantially exposed to various natural hazards (floods, droughts, cyclones, landslides, earthquakes, cyclones, and volcanic eruptions) with 89% financial losses from at least one category of natural disaster, and this percentage is still increasing [11].

India is amongst the top 10 countries experiencing disaster losses from 1998 to 2017 [12]. Recent statistics indicate that Jammu and Kashmir ranks as the third most vulnerable state in the Himalayan region, with an economic loss of $1 billion during the 2014 floods [13,14]. In addition, Jammu and Kashmir experienced a range of other disasters, including earthquakes, hailstorms, and human and livestock epidemics, at times escalating into disaster-like scenarios, resulting in substantial human and animal casualties, as well as damage to residences, public infrastructure, and crops [15]. Therefore, vulnerability assessment as an inherent characteristic is vital to gain insights into the different vulnerabilities of the communities dwelling in these fragile ecosystems, to mitigate potential harm from future extremities and stressors [16]. Inherent vulnerability is described as the evaluation of the susceptibility of any system to vulnerability based on intrinsic preconditions [17]. Vulnerability was theorized as a function of three components: exposure, sensitivity, and adaptive capacity [18]. Vulnerability assessment is generally carried out to recognize the hotspots characterized by low adaptive capacity and high sensitivity, with the aim of providing a deeper insight into the underlying drivers. The concept of vulnerability can further be divided into physical vulnerability, which considers factors such as access to basic amenities, healthcare, and social vulnerability, which is influenced by factors like gender, access to resources, and social marginalization [19–23].

Numerous studies have been conducted on vulnerability evaluation by means of indicator-based approach for social, and physical vulnerabilities [24–27]. A composite Climate Vulnerability Index (CVI) has been used to investigate and compare climate change vulnerability, considering dimensions such as adaptive capacity, sensitivity, and exposure [28]. While exploring diverse climate change scenarios, a framework has been developed for the analysis of household and village-level resilience and vulnerability in the Bhagirathi Basin of the Indian Western Himalayan region [29]. Ecological vulnerability has been studied on a spatio-temporal scale for the Upper Mzingwane sub-catchment of Zimbabwe by [30]. Furthermore, an indicator-based approach has also been utilized to assess the dynamics of vulnerability dimensions—exposure, sensitivity, and adaptive capacity—for framing a composite vulnerability index for districts of Maharashtra [31]. In a separate study, vulnerability to climate change-related health hazards has been carried out by employing geospatial indicators to evaluate neighbourhood variations [32]. In addition, studies have been conducted for inherent vulnerability based on social and ecological indicators at the district level for Indian states, revealing that 45% and 42% of districts are categorized as having medium and high vulnerability, respectively, exceeding the global average [5,16,33].

Jammu City, located in the Western Himalayas in a transitional zone between the Himalayan range and the plains, is susceptible to various hazards. In addition, during the past few years, the city has experienced rapid unplanned urbanization, development activities making it more vulnerable to any extremities in future [34]. Therefore, analysing inherent vulnerability is critical for combating the adverse impacts in future. The vulnera-

bility assessment has been carried out at broader scales, such as regional levels, but there is limited research that specifically examines and identifies the unique characteristics and vulnerabilities at city level. The present study has been carried out for inherent vulnerability at ward level in Jammu City using 22 indicators (social, ecological, infrastructural, and technological) due to the lack of comprehensive studies specifically aiming at understanding and assessing the vulnerability of individual wards within the city. This study will help in better resource management, besides enabling evidence-based decision-making for sustainable development, disaster risk reduction, and resilience-building efforts at the grassroots level.

## 2. Study Area

Jammu City is located in the south-western part of the Union Territory of Jammu and Kashmir. The area of Jammu Municipal wards extracted in the GIS environment was 72 sq.km, situated between 32°27′ N and 33°50′ N latitude and 75°19′ E and 75°20′ E longitude. The altitude ranges from 277 m to 520 m above sea level. Jammu experiences a warm and humid monsoon season from July to September, followed by a dry and chilly season from October to December. The city has a predominantly subtropical climate with noticeable seasonal variations. Known as the "city of temples", it holds significance due to its cultural heritage, geographical location, and historical connections (Figure 1). As per the 2011 census, Jammu City has a population of 5.1 lakh residents, while the Jammu Planning Area had 12.8 lakh residents in 2011, increasing to 14.9 lakh residents in 2019, with a literacy rate of 88.70% (Figure 2) [35]. The city centre includes densely populated areas such as Rehari, Ambphalla, Raghunath Bazar, and Mubarak Mandi. The development pattern in the main arteries of the city shows a dispersed nature compared to the density distribution in the surrounding districts. Jammu City is situated on the banks of the Tawi River, a tributary of the Chenab River, which originates from the Pir Panjal Range and provides water to the city. The city's landscape consists of plains, hills, and rivers. The plains are fertile and suitable for agriculture, stretching along the Tawi River. The city is surrounded by hills, including the Shivalik Range, which is part of the Himalayas.

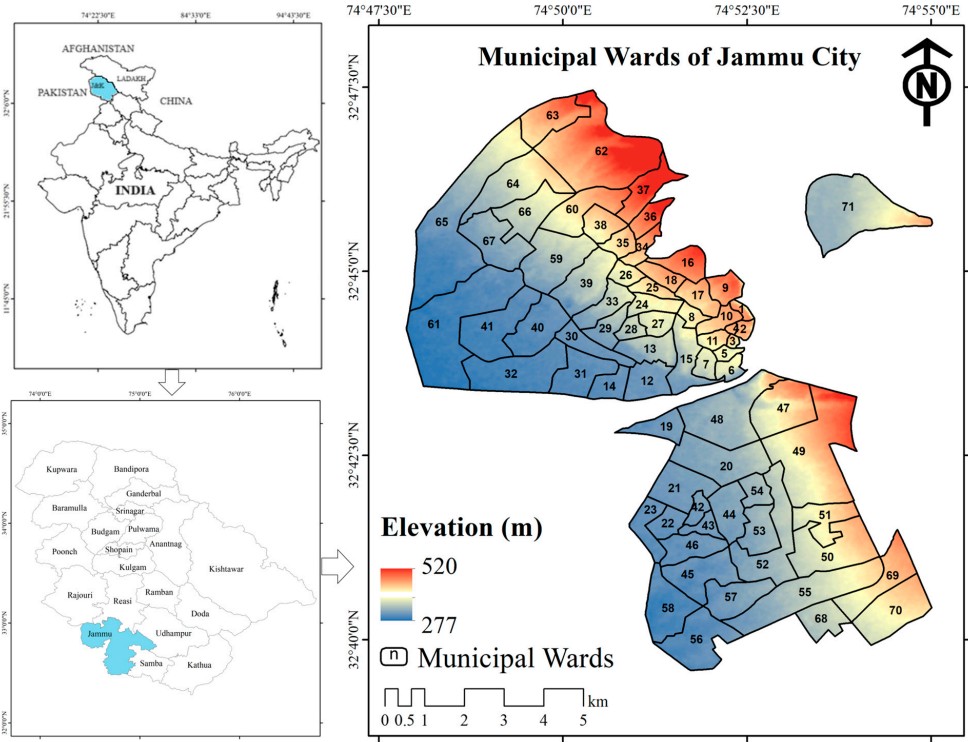

**Figure 1.** Jammu City map. created using municipal ward boundaries acquired from the Jammu Municipal Corporation overlaid on a Digital Elevation Model (DEM).

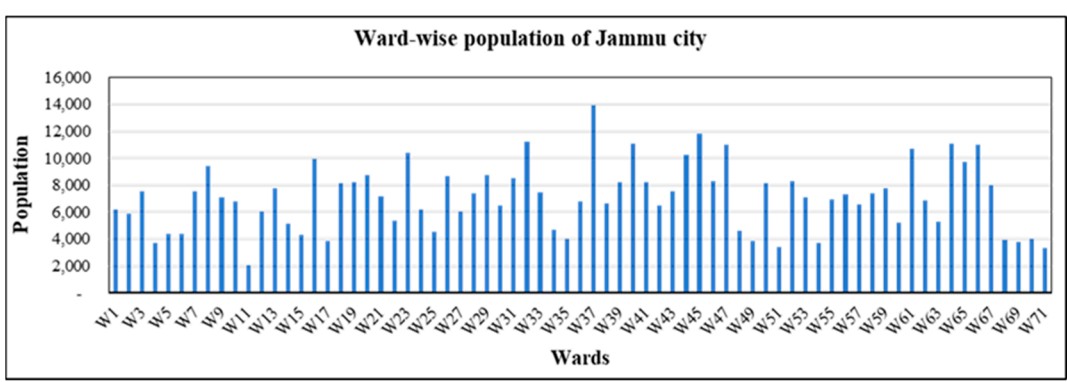

**Figure 2.** Population of Jammu City by ward (2011).

## 3. Materials and Methods

### 3.1. Materials

The present study utilized freely available datasets to analyse vulnerability in Jammu City at ward level. The primary data source was remote sensing data from Landsat 8 OLI and TIRS (acquired on 9 June 2021) from United States Geological Survey (USGS), with a 30 m spatial resolution. The images, acquired during the summer months, were employed to investigate the ecological indicators such as Land Surface Temperature (LST), land use/land cover and vegetation cover, and the Normalized Difference Vegetation Index (NDVI). The socio-economic data pertaining to the social, technological, and infrastructural components acquired from the Census of India and District Census Handbook for the year 2011 have been used for components of vulnerability, i.e., exposure, sensitivity, and adaptive capacity.

### 3.2. Methods

#### 3.2.1. Derivation of Ecological Indicators

The three major steps performed to derive the satellite-based ecological indicators include firstly, the classification of land use/land cover for the year 2021, which was carried out using the supervised classification method, utilizing a prior knowledge of the study area [36].

The classification process utilized a maximum-likelihood approach with equal prior probability. Attribute values were assigned based on cell probability for each class [37]. Ground validation was conducted to obtain field evidence of the several mapped land cover classes and establish correlations with corresponding image features. Finally, a GIS environment was utilized to calculate the area coverage of visually perceived Land Use and Land Cover (LULC) classes. Secondly, the normalized difference vegetation index (NDVI) was derived using the visible and infrared bands to identify the areas under high vegetation cover; thirdly, Land Surface Temperature (LST) was derived by converting Digital Numbers (DNs) of thermal bands to spectral radiance, followed by the computation of brightness temperatures; finally, estimation of land emissivity was carried out using the NDVI with subsequent conversion of brightness temperatures to Land Surface Temperature (LST) [38,39].

#### 3.2.2. Selection of Indicators

The vulnerability index for Jammu municipal wards has been derived as a function of exposure, sensitivity, and adaptive capacity. To derive the inherent vulnerability, indicators were finalized based on literature review, expert knowledge, and availability of datasets. Indicators serve as a proxy representation of affecting factors that may impact the outcome of a disaster [5].

A set of 22 indictors were analysed, grouped into four major categories, viz., ecological, social, infrastructure, and technological, for ward-level assessment. The indicators were

further divided into three components: adaptive capacity (literacy population, home ownership, household assets, pucca and semi-pucca structures, sanitation, electricity, safe drinking water, NDVI, health institutions), sensitivity (very young population, female population, SC/ST population, illiterate population, dependency on agriculture, non-workers, built-up, kacha structures, average size of household), and exposure (population, LST, population density, area of ward), keeping in view systems' susceptibility to the adverse effects, if any. The process of arriving at the inherent vulnerability index is given in Figure 3, and the detailed workflow under each component of the vulnerability model and the rationale for selection are presented in Figure 4 and Table 1, respectively.

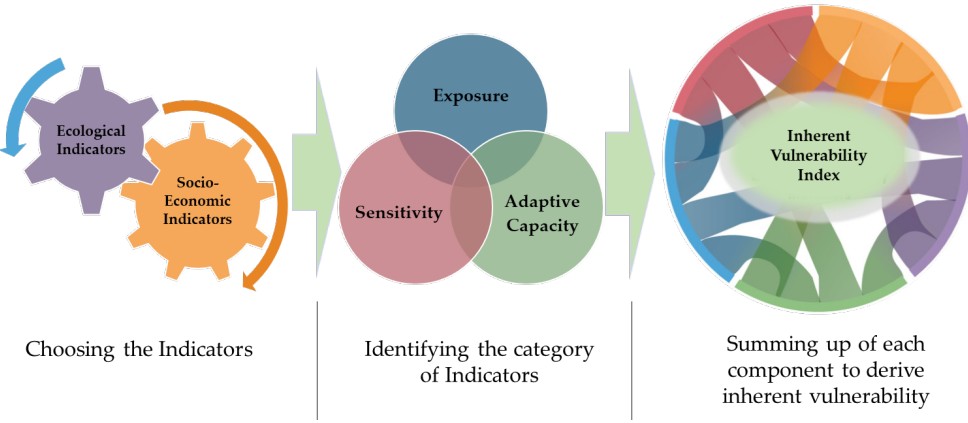

**Figure 3.** Step-wise process for developing the inherent vulnerability index.

**Table 1.** Indicators under each component for vulnerability assessment and rationale for selection.

| Indicators | Dimension | Functional Relationship with Vulnerability | Rationale for Selection | Source Data | References |
|---|---|---|---|---|---|
| Population | Exposure | Positive | Higher population increases vulnerability to hazards. | Census of India | [40,41] |
| LST | | | LST is a common indicator for heat exposure, and the increasing intensity and frequency of heatwaves pose significant environmental and health concerns. | USGS | [42] |
| Population density | | | High population density indicates that more people are vulnerable to any hazard. | Census of India | [43] |
| Area of Ward | | | The ward area serves as an exposure variable to assess the extent of risk in any type of extreme event. | Municipal Boundary | [44] |
| Very Young Population | Sensitivity | Positive | Children under 6 years are vulnerable to health, dependency, and disasters. | District Census Handbook | [45,46] |
| Female Population | | | Females, particularly those who are pregnant, are more vulnerable to any form of threat than men. | District Census Handbook | [47,48] |
| SC and ST Population | | | Due to historical discrimination, social exclusion, economic inequality, and limited access to opportunities, Scheduled Castes (SC) and Scheduled Tribes (ST) are particularly vulnerable, demanding targeted interventions for equitable development. | District Census Handbook | [27,49] |
| Illiterate Population | | | Illiterate persons are susceptible to any form of risk occurrence, because they are unaware of the potential dangers associated with these events. | District Census Handbook | [50,51] |
| Dependency on Agriculture | | | The agriculture sector as a source of livelihood is extremely sensitive to vagaries of weather and any variability can lead to loss of productivity, affecting livelihood and food security. | District Census Handbook | [52,53] |

**Table 1.** *Cont.*

| Indicators | Dimension | Functional Relationship with Vulnerability | Rationale for Selection | Source Data | References |
|---|---|---|---|---|---|
| Non-Workers | Sensitivity | Positive | People's unemployment status is linked to poverty, making them vulnerable. | District Census Handbook | [54,55] |
| Kacha Structures | | | In the absence of a better-quality house, households may face increased vulnerability to the risks posed by extreme conditions, considering its role as an indicator of socioeconomic status. | District Census Handbook | [56,57] |
| Average size of Household | | | Greater household size indicates more social diversity and potentially varying vulnerability levels, especially in densely populated urban areas. This can increase demand for critical infrastructure and services, straining them during and after hazard events. | District Census Handbook | [58,59] |
| Built-up | | | Densely built-up areas and infrastructure with limited space and escapes routes lead to vulnerability. | USGS | [56,60] |
| Literacy Population | Adaptive Capacity | Negative | Literate people have a negative relation with Vulnerability; because of their enhanced knowledge and abilities, educated people confront risks and challenges. | District Census Handbook | [13,27] |
| Home Ownership | | | Homeowners are less vulnerable due to increased stability, security, and asset protection. | District Census Handbook | [61,62] |
| Household assets | | | Households owning assets such as a car, a bicycle, a banking facility, internet access, and other appliances like a computer are considered less vulnerable due to increased mobility, financial inclusion, connectivity, and access to information and resources, contributing to overall resilience. | District Census Handbook | [63,64] |
| Pucca and Semi-pucca structures | | | Possession of a better–quality house will improve the capacity of households to withstand the risks from any extreme conditions. | District Census Handbook | [56,65] |
| Sanitation | | | Sanitary amenities, including bathrooms and efficient waste management, contribute to community health. Robust individual health enhances resilience to environmental shocks, facilitating adept adjustment to evolving conditions. | District Census Handbook | [66,67] |
| Electricity | | | Access to electricity allows better functioning of television, radios, mobile phones, and modern technologies, thereby improving access to relevant information which enables people to make informed decisions and to take proactive adaptation measures against any risk. | District Census Handbook | [65,68–70] |
| Safe drinking water | | | Water is a critical resource both for domestic consumption and agriculture. Hence, access to water resources and the quality of water for household use is one of the important assets in any community. | District Census Handbook | [71,72] |
| Health institution | | | Access to functional healthcare infrastructure is essential for the overall health and well-being of any community. | District Census Handbook | [44,73,74] |
| NDVI | | | Green cover in cities and towns serves to balance temperatures, counteracting urban heat, and providing a mitigation effect on the heat environment. The absence of green space can elevate the risk of higher heat exposure. | USGS | [75,76] |

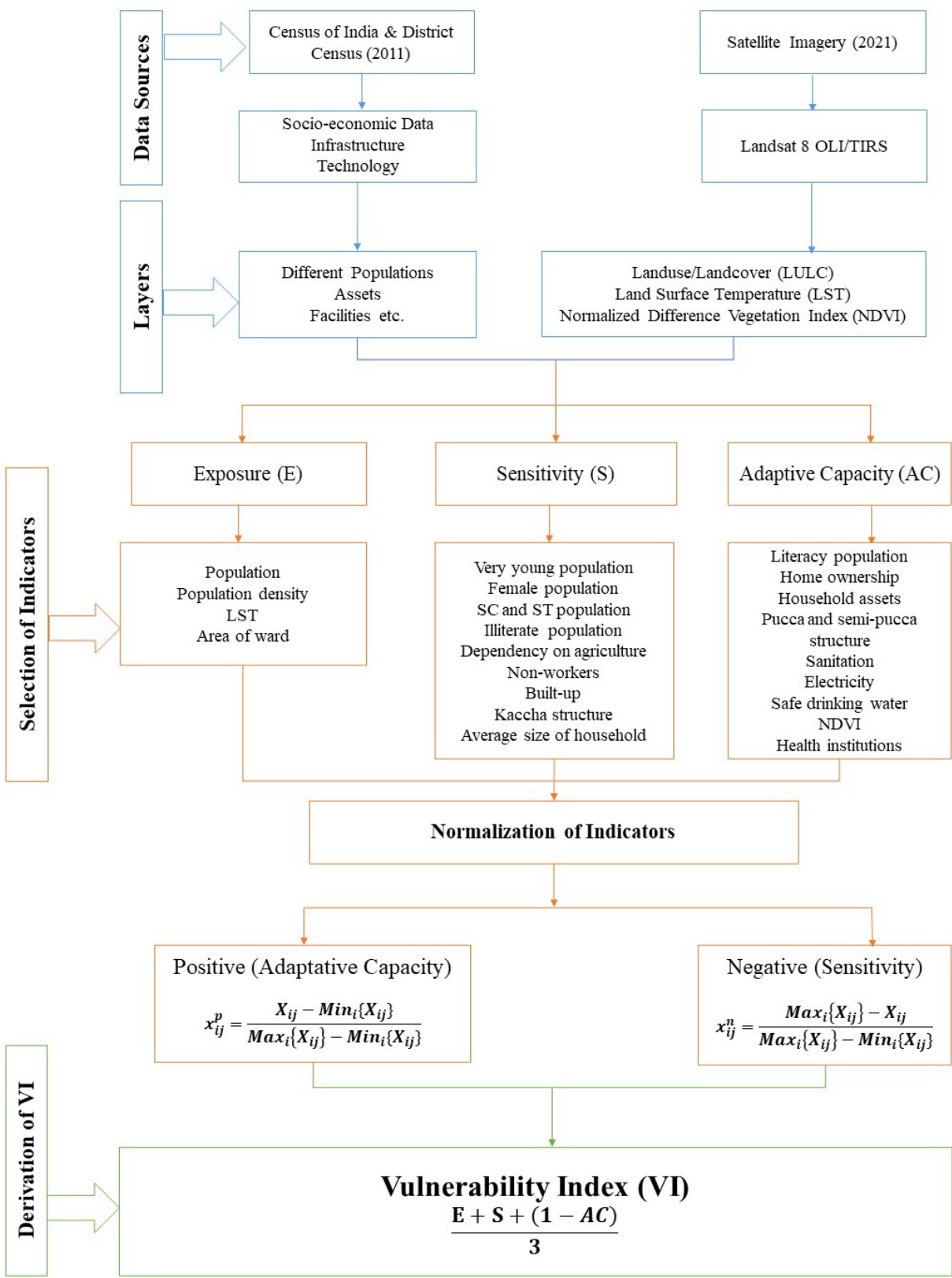

**Figure 4.** Flow-chart depicting indicators used, with detailed methodology adopted.

3.2.3. Calculation of the Vulnerability Index

The vulnerability index has been calculated utilizing an indicator-based approach, as these methods have found widespread application for comprehending and visualizing

vulnerability in various regions and contexts [17,77–85]. The calculation of the vulnerability index involves four major steps, viz., data segregation, processing of data, computation of the index, and reclassification. During the initial phase of data processing, all indicators were systematically classified into three dimensions, Exposure, Sensitivity, and Adaptive Capacity, throughout the entire city. The subsequent step included establishing the functional association of each indicator with vulnerability, identifying whether it had a positive or negative impact on vulnerability. After data categorization, a standardization process was implemented by converting all data into percentage values. Following this, normalization was carried out on the percentage values within each dimension, resulting in dimensionless units for 22 indicators, respectively. This normalization process was conducted at the ward level, guided by the functional relationship of the indicators with vulnerability. For indicators positively related to vulnerability, i.e., where vulnerability increases with an increase in the indicator value, the following formula was employed:

$$x_{ij}^{p} = \frac{X_{ij} - Min_i\{X_{ij}\}}{Max_i\{X_{ij}\} - Min_i\{X_{ij}\}} \tag{1}$$

For negatively related indicators, i.e., where vulnerability decreases with an increase in the value of the indicator, the following formula was used:

$$x_{ij}^{n} = \frac{Max_i\{X_{ij}\} - X_{ij}}{Max_i\{X_{ij}\} - Min_i\{X_{ij}\}} \tag{2}$$

where $X_{ij}$ is the value of $j$th indicator for $i$th indicator, $Min_i \{X_{ij}\}$ is the minimum and $Max_i \{X_{ij}\}$ is the maximum value of the $j$th indicator. $x_{ij}^{p}$ and $x_{ij}^{n}$ are the normalized values for positively and negatively related indicators, respectively, which lie between 0 and 1 [86,87]. The value 1 will correspond to maximum vulnerability, whereas 0 will correspond to a ward with minimum vulnerability with respect to a particular indicator [86].

Upon calculation of normalized values, contributing factors were aggregated and the vulnerability index was calculated using Equation (3) and outputs were reclassified into five categories for visualization and area calculation [74].

$$VI = \frac{E + S + (1 - AC)}{3} \tag{3}$$

where *VI* represents the Inherent Vulnerability Index, Av (*E*, *S*) represent the average of Exposure and Sensitivity, and *AC* means Adaptive Capacity.

## 4. Results & Discussion

### *4.1. Land Use and Land Cover (LULC)*

The LULC mapped for Jammu City using Landsat 8 OLI satellite imagery depicted five different categories of LULC, viz., agriculture, built-up land, forest, waterbodies, and wasteland. It was observed that built-up land occupied most of the area in the year 2021, i.e., 64.5 sq.km, which corresponds to 89.5% of the total area of municipal wards (Table 2 and Figure 5). The incessant transformation of agricultural land to built-up land over the past years. mainly in response to the demands of the main city, has resulted in the dominance of built-up land in the region. The city has exhibited distinctive patterns in its population growth, specifically, between 1971 and 1981, and 1981 to 2001, when population increased by more than double, attributed to mass migration from the valley in the 1990s. Additionally, demographic changes have been influenced significantly by urbanization and industrialization [88]. The other land use categories, such as agriculture and forest, occupy an area of 5 sq.km and 1.8 sq.km, respectively. From the analysis, it was seen that the minor types of land use in Jammu City correspond to waterbodies and wasteland, covering an area of 0.3 sq.km. The LULC was overlaid with the municipal ward boundaries of Jammu City to calculate the area covered by each LULC within the 71 wards. It was observed that

out of 71 wards natural features like forest, agriculture, and waterbody are present in ward number 15, 11, and 10 respectively, whereas the built-up category is dominant in all wards (Figure 6).

**Table 2.** LULC statistics for Jammu City for the year 2021.

| Land Use Classes | 2021 (Area) | |
|---|---|---|
| | **sq.km** | **(%)** |
| Built up | 64.5 | 89.5 |
| Agriculture | 5 | 7 |
| Forest | 1.8 | 2.5 |
| Water bodies | 0.3 | 0.5 |
| Wasteland | 0.3 | 0.5 |
| Total | 72 | 100 |

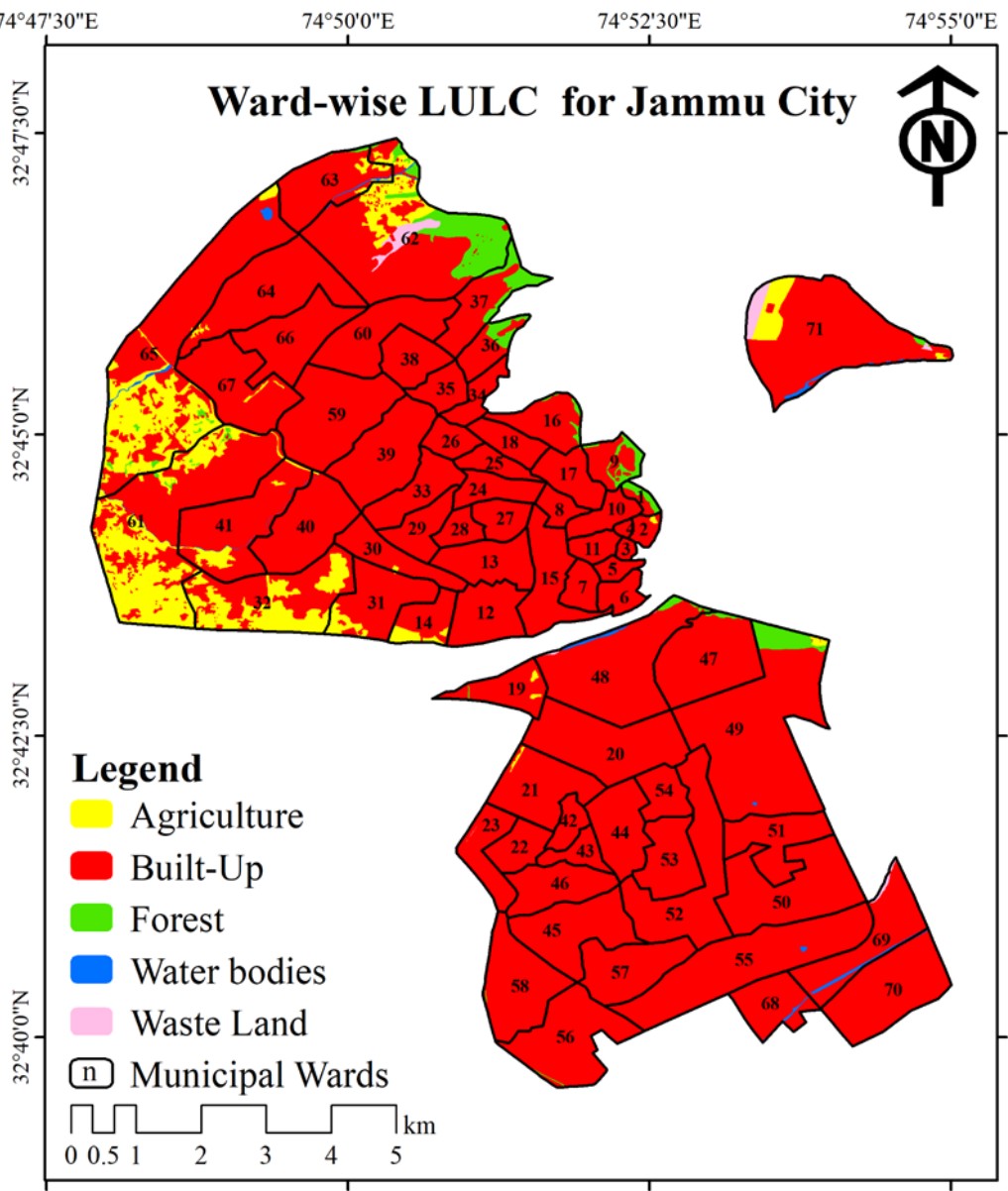

**Figure 5.** Land use/land cover map of Jammu City overlaid on municipal ward boundaries.

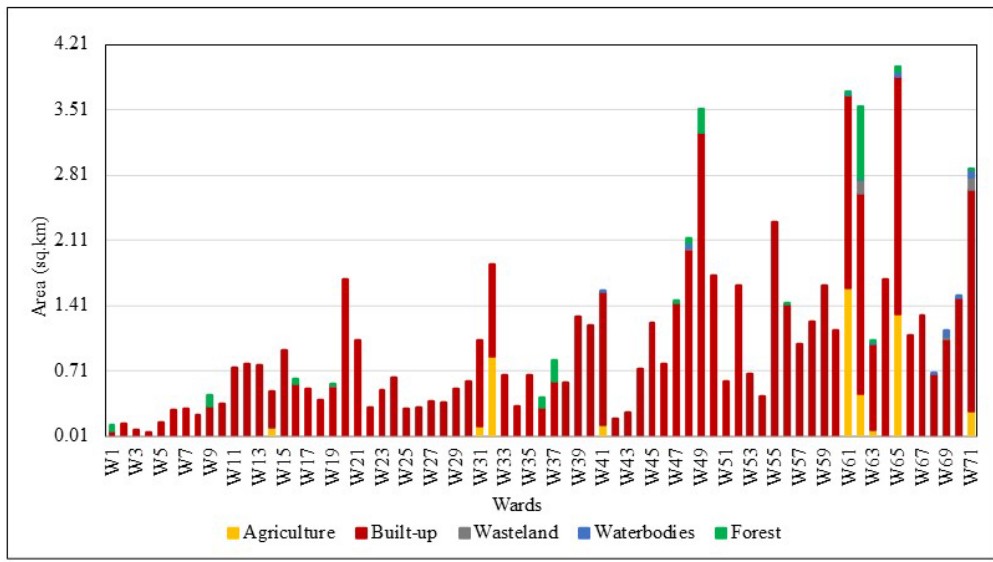

**Figure 6.** Distribution of LULC classes in Jammu City by ward.

*4.2. Normalized Difference Vegetation Index (NDVI)*

When identifying changes in land use and cover brought about by urban and economic development, the NDVI index is a crucial tool. A region's NDVI is significantly impacted by changes in LULC [89]. The NDVI map's extracted values can be used to identify the effective factors affecting land cover [90]. The Normalized Difference Vegetation Index (NDVI) derived from Landsat 8 OLI exhibits variability, with a maximum and minimum NDVI of 0.48 and −0.04, respectively. The mean and standard deviation observed were 0.15 and 0.05, respectively. The NDVI was classified into five classes, viz., very low, low, moderate, high, and very high, for the statistical calculations presented in Table 3 and Figure 7.

**Table 3.** NDVI statistics for Jammu City for the year 2021.

| NDVI | Range | Area | |
|---|---|---|---|
| | | sq.km | % |
| Very Low | −0.04–0.10 | 15.8 | 22.0 |
| Low | 0.1–0.14 | 20.8 | 28.8 |
| Moderate | 0.14–0.18 | 19.0 | 26.4 |
| High | 0.18–0.23 | 12.6 | 17.6 |
| Very High | 0.23–0.48 | 3.8 | 5.2 |
| Total | | 72.0 | 100.0 |

From the statistical analysis, it was observed that the majority of municipal wards fall into the low category of NDVI, comprising 20.8 sq.km (28.8%) of the total study area, followed by moderate, 19.0 sq.km (26.4%). The domination of low NDVI in the study area is due to the presence of high built-up area (89.5%). It was perceived that there was a negative correlation between NDVI and built-up areas, but a positive correlation between LULC and NDVI in terms of vegetation. These findings also aligned with findings of other researchers [91–93]. The alterations in land use and land cover over both short and extended time frames are predominantly attributed to urbanization and population growth. For instance, a study revealed that over the past three decades changes in land use and land cover (LULC) have led to multiple declines in the Normalized Difference Vegetation Index (NDVI) within the metropolitan areas of the Lower Himalayan region [94]. This decline was primarily attributed to the substitution of vegetation with impermeable surfaces [95]. The high and very high categories cover 16.4 sq.km, which is 22.8% of the total area of study

under consideration. The results of cross tabulation carried out for NDVI and 71 wards are given in Figure 8.

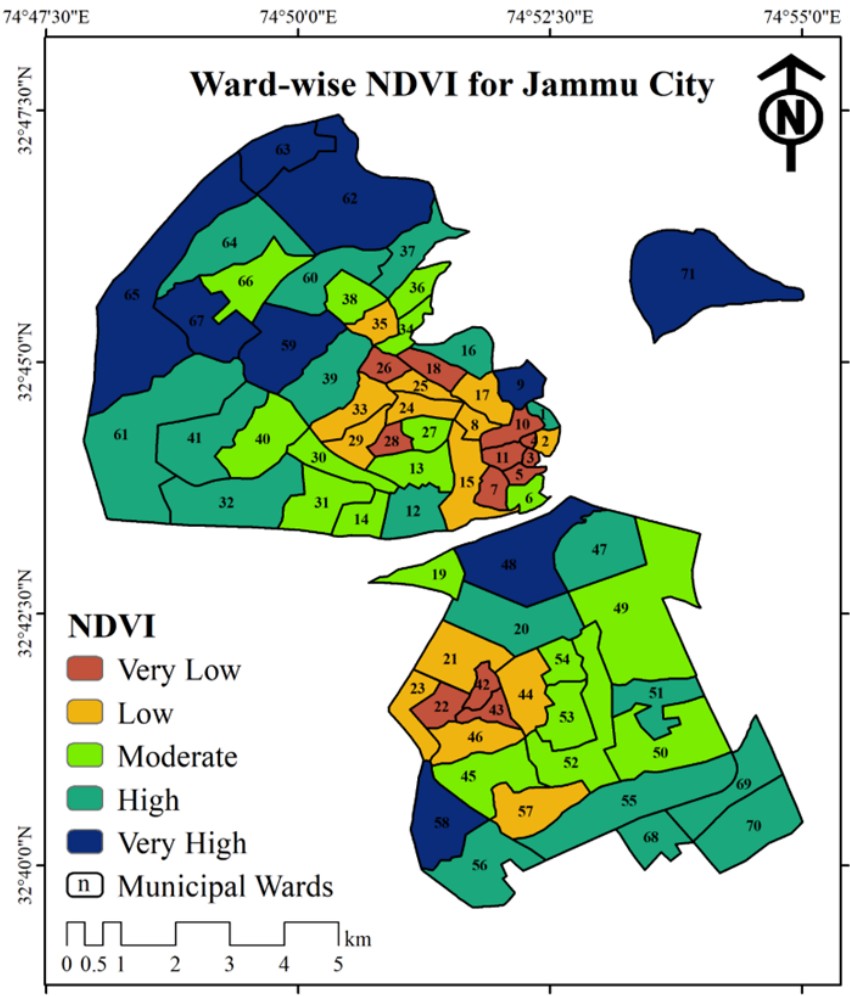

**Figure 7.** NDVI map of Jammu City overlaid on municipal ward boundaries.

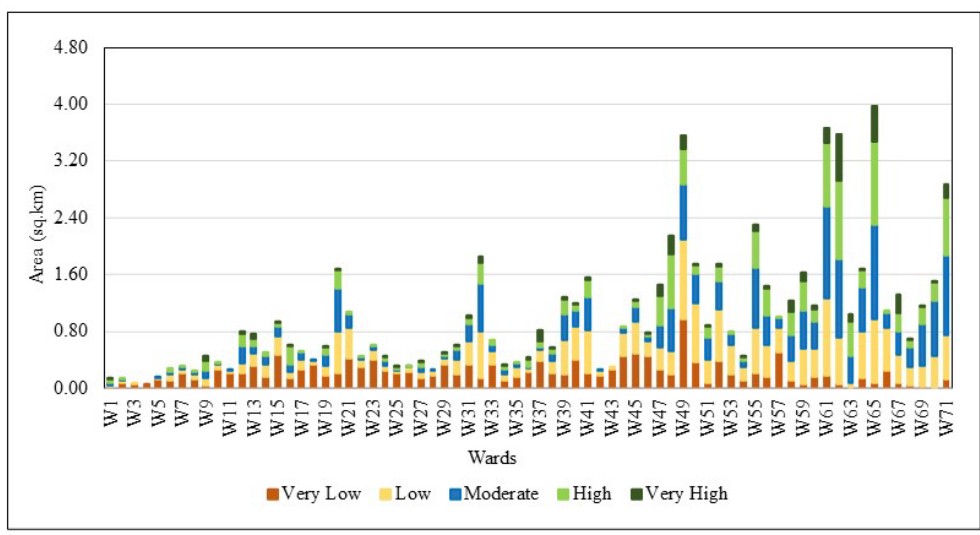

**Figure 8.** Ward-wise distribution of different categories of NDVI in Jammu City.

It was observed that the very high category of NDVI, ranging from 0.23–0.48, which corresponds to presence of high vegetation, covers a total area of 3.8 sq.km. Out of this,

wards W62(Chinore/Keran-1) and W48(Bahu West) cover the majority of the area, i.e., 0.64 sq.km and 0.25 sq.km, respectively. The high NDVI category, ranging from 0.18–0.23, covered the maximum area out of 12.6 sq.km in wards W65(Barnai/Upper Dharmal) and W61(Patta Paloura), covering 3.98 sq.km and 3.67 sq.km, respectively. The moderate category, which ranges from 0.14–0.18, exhibits maximum coverages of 0.98 sq.km and 0.88 sq.km in wards W15 (Partap Ghar), and W51 (Channi Himmat/Thanger Area). The very low category of NDVI which ranges from −0.04–0.10, corresponding to low vegetation cover, exhibits maximum area in the wards, i.e., 15.8 sq.km covering 22% of the total area of wards. The highest area in the very low category of NDVI is exhibited in wards W4 (Fattu Choghan, and W3 (Mast Garh, encompassing a total area of 0.98 sq.km.

### 4.3. Land Surface Temperature (LST)

LST serves as a vital variable for describing the spatial-temporal patterns of near-surface air temperature in hilly terrains because of the presence of typically fewer meteorological stations [96,97]. Therefore, LST was taken as a proxy for temperature and one of the indicators for the exposure dimension. The highest and lowest LSTs recorded were 37.32 °C and 29.05 °C, respectively (Figure 9), with the mean being 33.99 °C and a standard deviation of 0.75. For statistical calculations, the LST was classified into five classes: very low, low, moderate, high, and very high, as shown in Table 4. According to the statistical analysis, the majority of municipal wards fall into the moderate (33.53–34.18) category, accounting for 27.5 sq.km (38.2%) of the entire study area, followed by the high category, with sq.km (33.3%) in the high (34.18–34.96) category. The presence of high built-up (89.5%) accounts for the dominance of high LST in the study area. The high value of LST (mean temperature of 35.72 °C) was found in Gangyal ward (W57), the main industrial estate of Jammu City surrounded by residential-cum-commercial built-up land, since both types have a direct association with land surface temperature [98]. In contrast, Chinore (W62) and Janipur West (W32) had mean temperatures of 33.33 °C and 32.96 °C, respectively, due to neighbouring forest terrain. Areas with open vegetation show moderate values of LST, as urban vegetation and LST are negatively correlated [99–101]. The results of cross-tabulation carried out for LST and 71 Wards are given in Figure 10. It was observed that the very high category of LST ranging from 34.96–35.32 °C, corresponding to the presence of built-up land, covers a total area of 4.8 sq.km. Out of this, wards W57, Gangyal-2, and W4,; Narwal/Channi Rama, cover the majority of area, i.e., 0.82 sq.km and 0.75 sq.km, respectively. In the high LST category, ranging from 34.18–34.96 °C, wards W49, Narwal/Channi Rama, and W45; Digiana, covered the maximum area of 1.85 sq.km and 0.82 sq.km, respectively. Wards W61, Patta Paloura, and W62, Chinore/Keran-2, exhibit the maximum coverages of 1.56 sq.km and 1.47 sq.km in the moderate category. The highest area in the very low category of LST is exhibited in wards W62, Chinore/Keran-1, and W65; Barnai/Upper Dharmal, encompassing a total area of 1.09 sq.km. The presence of moderate to high categories of LST in the study area is due to the substitution of the rural land cover type with concrete urban elements, which may result in increased surface runoff, reduced evapotranspiration, and lower air and water quality [102]. Therefore, wards characterized by higher LST in areas with low vegetation and significant urban development may result in increased surface runoff and reduction in ground water recharge [103,104].

**Table 4.** LST statistics for Jammu City for the year 2021.

| LST | Range (°C) | Area | |
|-----|------------|------|---|
| | | sq.km | % |
| Very Low | 32.96–33.46 | 2.8 | 3.9 |
| Low | 33.62–34.05 | 12.9 | 17.9 |
| Moderate | 34.05–34.42 | 27.5 | 38.2 |
| High | 34.42–34.87 | 24.0 | 33.3 |
| Very High | 34.87–35.72 | 4.8 | 6.7 |
| Total | | 72.0 | 100.0 |

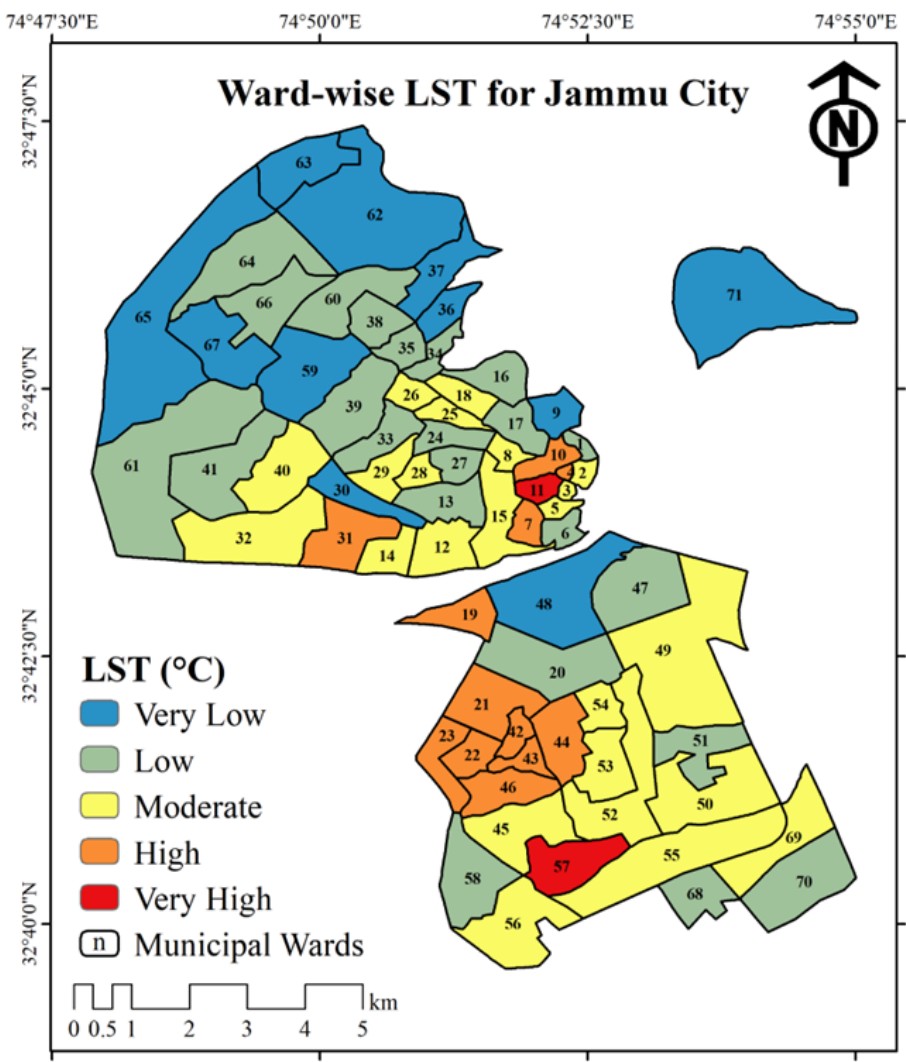

**Figure 9.** LST map of Jammu City overlaid on municipal ward boundaries.

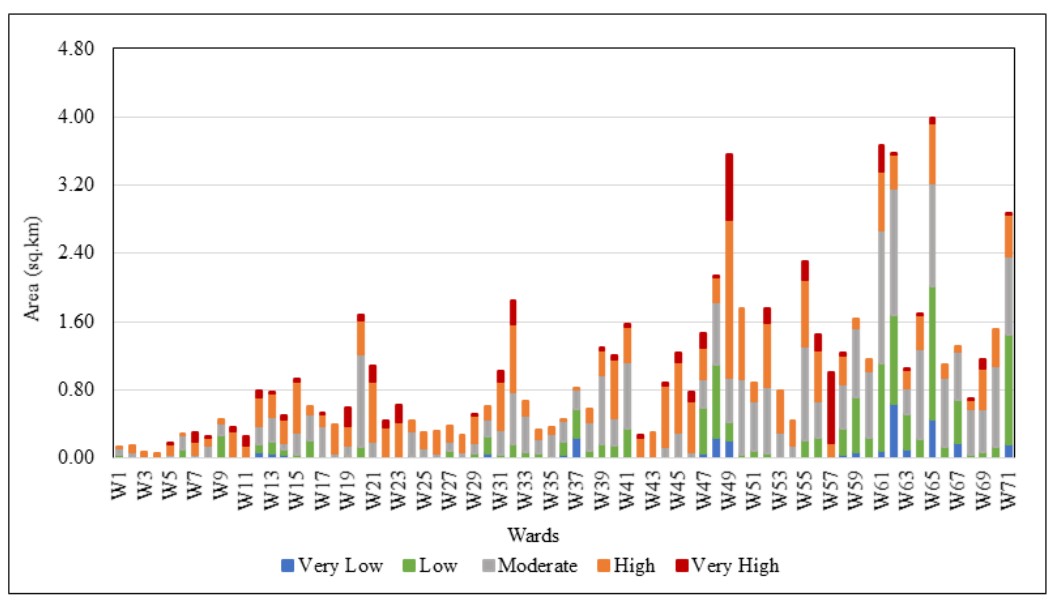

**Figure 10.** Distribution of LST classes in Jammu City by ward.

*4.4. Spatial Distribution of Variables of Vulnerability*

4.4.1. Exposure (E)

The evaluation of exposure variables utilized the normalized extraction method, focusing on four selected indicators: population, population density, Land Surface Temperature (LST), and ward area. The significance of considering both population and the resultant population density as integral elements in assessing exposure have been emphasized [105,106].

The dominant contributors to high exposure were identified as population (33.69%) and LST (32.96%). Large urban populations with significant social stratification mean that it can contribute to increased vulnerability to any kind of extreme event within specific urban subsets [107,108]. Additionally, since a higher LST implies that a population has been exposed to excessive temperatures, replacing rural land cover types with concrete urban features will have an impact on vulnerability, which in turn will have resultant impacts on human population [109]. Since the fringes of Jammu City are categorized as having large ward sizes, they contribute little to exposure. But people who live in highly urbanized environments often migrate to these open spaces, which puts further strain on the land [110]. In contrast, population density (13.87%) and ward area (19.46%) demonstrated relatively smaller influences on exposure. The exposure levels were categorized into five tiers: very low, low, moderate, high, and very high.

Results from the statistical analysis revealed that the majority of wards come under the high exposure category, i.e., 29.39 sq.km (40.78%), followed by the low exposure category, covering 14.17 sq.km (19.66%) (Table 5 and Figure 11). Wards with very low exposure constituted 5.26 sq.km (7.29%), while those with moderate exposure covered 11.89 sq.km, and areas classified as under very high exposure comprised 11.36 sq.km of the total study area. The spatial distribution shows that wards 4 (Fattu Choghan), 23 (Nai Basti), 32(Gole Gujral), 45 (Digiana), 61 (Patta Paloura), and 65 (Barnai/Upper Dharmal) come under the very high category of exposure, and the most important contributing driver is population. In addition to population, LST and area of ward are contributing drivers in ward 23, 61 and 65.

**Table 5.** Area statistics of Exposure, Sensitivity, Adaptive Capacity and Vulnerability Index for Jammu City.

| Classes | Range (E) | Area (E) | | Range (S) | Area (S) | | Range (AC) | Area (AC) | | Range (VI) | Area (VI) | |
|---|---|---|---|---|---|---|---|---|---|---|---|---|
| | | sq.km | % | | sq.km | % | | sq.km | % | | sq.km | % |
| Very Low | 0.15–0.22 | 5.26 | 7.29 | 0.04–0.16 | 6.93 | 9.62 | 0.14–0.24 | 8.6 | 12.0 | 0.29–0.33 | 9.77 | 13.55 |
| Low | 0.22–0.28 | 14.17 | 19.66 | 0.16–0.24 | 11.85 | 16.45 | 0.24–0.33 | 27.3 | 37.9 | 0.33–0.39 | 11.35 | 15.75 |
| Moderate | 0.28–0.34 | 11.89 | 16.50 | 0.24–0.33 | 27.72 | 38.46 | 0.33–0.41 | 16.9 | 23.4 | 0.39–0.42 | 14.42 | 20.00 |
| High | 0.34–0.43 | 29.39 | 40.78 | 0.33–0.46 | 22.88 | 31.75 | 0.41–0.54 | 12.4 | 17.3 | 0.42–0.48 | 18.38 | 25.49 |
| Very High | 0.43–0.55 | 11.36 | 15.76 | 0.46–0.63 | 2.69 | 3.73 | 0.54–0.64 | 6.8 | 9.4 | 0.48–0.59 | 18.15 | 25.17 |
| Total | - | 72 | 100 | - | 72 | 100 | - | 72 | 100 | - | 72 | 100 |

Distinct wards exhibited diverse levels of exposure; notably, Wards 65 and 61 stood out with exceptionally high exposure, encompassing areas of 3.97 and 3.66 sq.km, respectively. The drivers which were responsible for very high exposure also contributed to the high exposure class in areas like Jullaka Mohalla, Poonch House, Bahu East, Sanjay Nagar, Gangyal, and Chak Changarwan. Low exposure is observed in 15 wards out of 71 wards, viz., W48 (Bahu West), W69 (Sainik Colony-1), W70 (Sainik Colony-2), W14 (Bhagwati Nagar), W33 (Shiv Nagar), W24 (Rehari Colony North), W27 (Bakshi Nagar), W11 (Mohalla Malhotrian), W5 (Talab-khatikan), W17 (Amphalla), W9 (Mohalla ustad), W38 (Paloura), W60 (Paloura Centre), W67(Lower Muthi), and W71 (Tawi Housing Sidhra/Vihar Cly), which is due to low population density. Out of a total of 15 wards in the low category, W71 and W48 exhibited high areas of 2.89 and 2.15 sq.km, respectively.

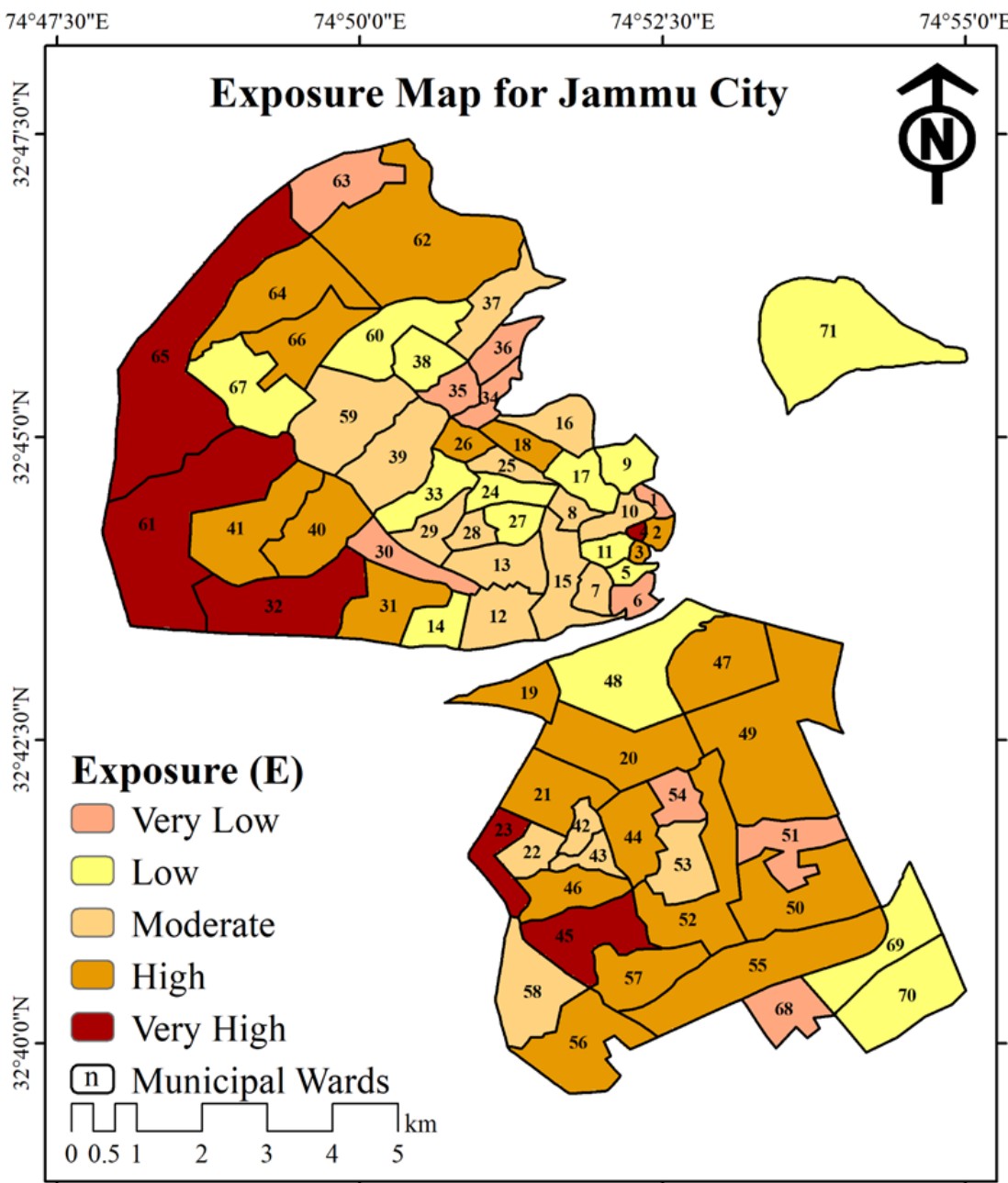

**Figure 11.** Spatial distribution of exposure map of Jammu City overlaid on municipal ward boundaries.

### 4.4.2. Sensitivity (S)

In assessing sensitivity variables, the normalized extraction method was applied, focusing on nine specific indicators: very young population (<6 years), illiterate population, non-workers, SC and ST population, dependency on agriculture, kacha structures, female population, built-up areas, and average household size. The prominent factors contributing to high sensitivity in the ward are the female population (15.76%), non-workers (15.05%), very young population (13.54%), illiterate population (10.61%), and built-up areas (9.7%). Additionally, the least contributing elements encompass the average size of households (8.31%), dependency on agriculture (6.24%), SC and ST population (5.83%), and kacha structures (4.93%). The sensitivity levels were classified into five categories: very low, low, moderate, high, and very high. The analysis indicated that the majority of wards fall within the moderate sensitivity category, encompassing 27.72 sq.km (38.46%). Following this, wards in the high sensitivity category covered 22.88 sq.km (31.75%) (Table 5 and Figure 12).

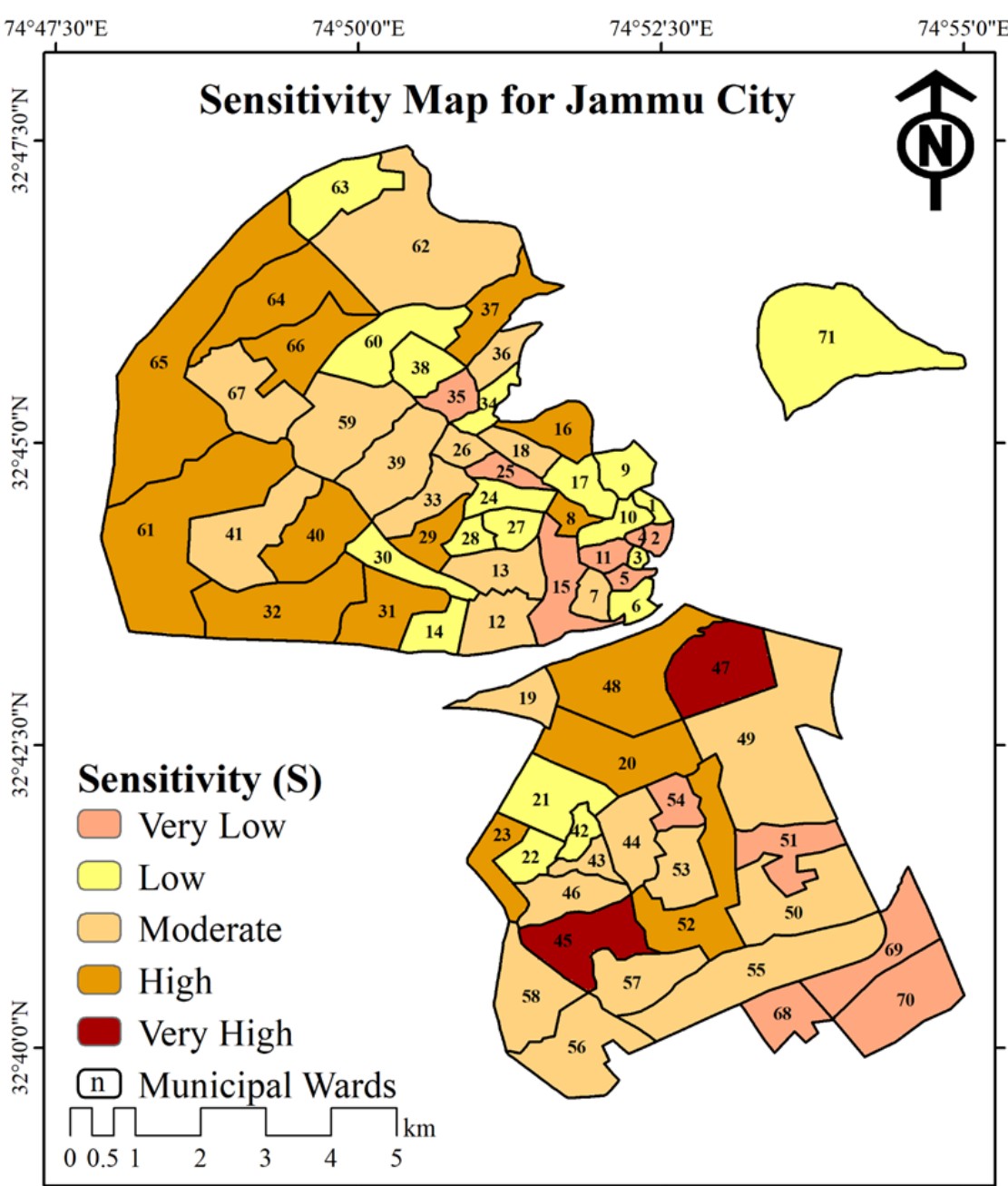

**Figure 12.** Spatial distribution of sensitivity map of Jammu City overlaid on municipal ward boundaries.

Areas characterized by very high sensitivity constituted 2.69 sq.km (3.73%), whereas those with low sensitivity covered 11.85 sq.km (16.45%). The regions classified as having very low sensitivity comprised 6.93 sq.km (9.62%) of the total area. The spatial distribution shows that wards W45 (Digiana) and W47 (Bahu East) come under the very high category of sensitivity. In W45, it was observed that the drivers of sensitivity are very young population, non-workers, dependency on agriculture and female population. However, these factors also contribute in W47, in addition to illiterate population and SC and ST population. The exception lies in the driver, i.e., dependency on agriculture, which does not contribute to sensitivity on the higher side in W47, besides its areal coverage of 1.46 sq.km. The wards which displayed high sensitivity in terms of high average value, is mainly due to drivers such as non-workers and female population in wards, viz., W32 (Gol Gujral), W61 (Patta Paloura), W64 (Chak Chagarwan), and W65 (Barnai/Upper Dharmal). In wards W48 (Bahu West) and W52 (Channi/Channi Bija), only one driver was found responsible for

higher sensitivity, i.e., kacha structures. The low sensitivity observed in 19 wards out of 71 wards is due to low dependency on agriculture, low SC and ST population, and fewer kacha structures.

It has been reported globally that women consistently face heightened risks compared to men in various types of disasters, both during the events and in the subsequent recovery periods [111]. Despite the common lack of gender-disaggregated data in disaster reporting, studies have indicated a significantly higher mortality rate among women compared to men as a result of disasters [112]. Women's increased vulnerability to the effects of natural disasters and climate change stems not only from biological and physiological differences but also notably from socioeconomic disparities and inequitable power relations [113].

The WHO has also reported that people at a low socio-economic level, women, and young people in any community are particularly susceptible to anxiety and mood problems associated with catastrophes [114]. Children are already more susceptible to the direct consequences of any extreme event, according to analysis, because they rely more on adult family members and social support systems, therefore increasing the system's sensitivity [115,116]. The results of this study are also in conformity with the findings that the portion of the population that is unemployed is vulnerable to the effects of climate change as a result of insufficient funds and increases the system's susceptibility [117,118].

### 4.4.3. Adaptive Capacity (AC)

The evaluation of adaptive capacity variables was calculated based on nine selected indicators such as literate population, homeowners, household assets, pucca and semi-pucca constructions, sanitation, electricity, safe drinking water, health institutions, and NDVI. Health institutions (25.75%), literate population (18.36%), family assets (15.82%), NDVI (13.98%), and home ownership (9.04%) contribute to strong adaptive capacity, whereas safe drinking water (2.82%), electricity (3.37%), pucca and semi-pucca dwellings (4.35%), and sanitation (6.5%) have a less significant contribution (Table 5). The adaptive capacity, categorized into five levels, revealed that the majority of wards come under the low category, covering an area of 27.3 sq.km (37.9%), followed by the moderate category, i.e., 16.9 sq.km (23.4%). The high adaptive capacity category covered an area of 12.4 sq.km (17.3%), whereas very high covered 6.8 sq.km (9.4%). followed by the low category, i.e., 8.6 sq.km (12%). The spatial distribution shows that wards W48 (Bahu West), W52 (Channi/Channi Bija), and W71 (Tawi Housing Sidra/Vihar City) come under the very high category of adaptive capacity, due to their better health facilities and sanitation (Figure 13). In addition, it was observed that W48 has higher adaptive capacity among all the three wards because of good literacy condition and presence of pucca and semi-pucca structures. From the analysis, it was perceived that very low adaptive capacity in wards is due to poor facilities in terms of household assets, health institutions, and literacy.

The Special Report on Extreme Events and Disasters by the Intergovernmental Panel on Climate Change (IPCC) in 2012 warned that climate change is poised to amplify the scale, frequency, duration, and geographical reach of weather-related events. This heightened impact poses more substantial challenges to the health of future populations [119]. Impacts of climate change can lead to premature mortality, heatstroke, and respiratory emergency room visits and hospitalization, as the presence of such a good healthcare network is essential to cope with the adversities of climate change [120]. According to research, numerous healthcare facilities were enlisted to offer refuge to individuals without electricity at home and to furnish essential supplies to other hospitals in the midst of climatic disasters [121].

Some studies have emphasized the necessity to enhance the emergency preparedness of healthcare facilities, underscoring the importance of enhancing disaster resilience to address the challenges posed by a shifting climate [122–124]. Therefore, the presence of a good number of health facilities around Jammu City, being the winter capital of Jammu and Kashmir, enhances the city's adaptive capacity. Moreover, a high literacy rate being a factor to increase the vulnerability aligns with the results of researchers who also reported that the increasing education levels and student–teacher enrolment ratio in the Nashik

and Pune regions of Maharashtra state helped in making the system adaptive [125]. It has been reported that reducing the incidence of malaria, promoting literacy rate, and access to electronic gadgets and other family assets hold the most promise to reduce vulnerability by improving adaptive capacity [123]. Another study also reported that areas having a lower literacy rate were unable to cope immediately from disasters, thereby underlining the importance of the literacy rate as an indicator for adaptive capacity [126,127].

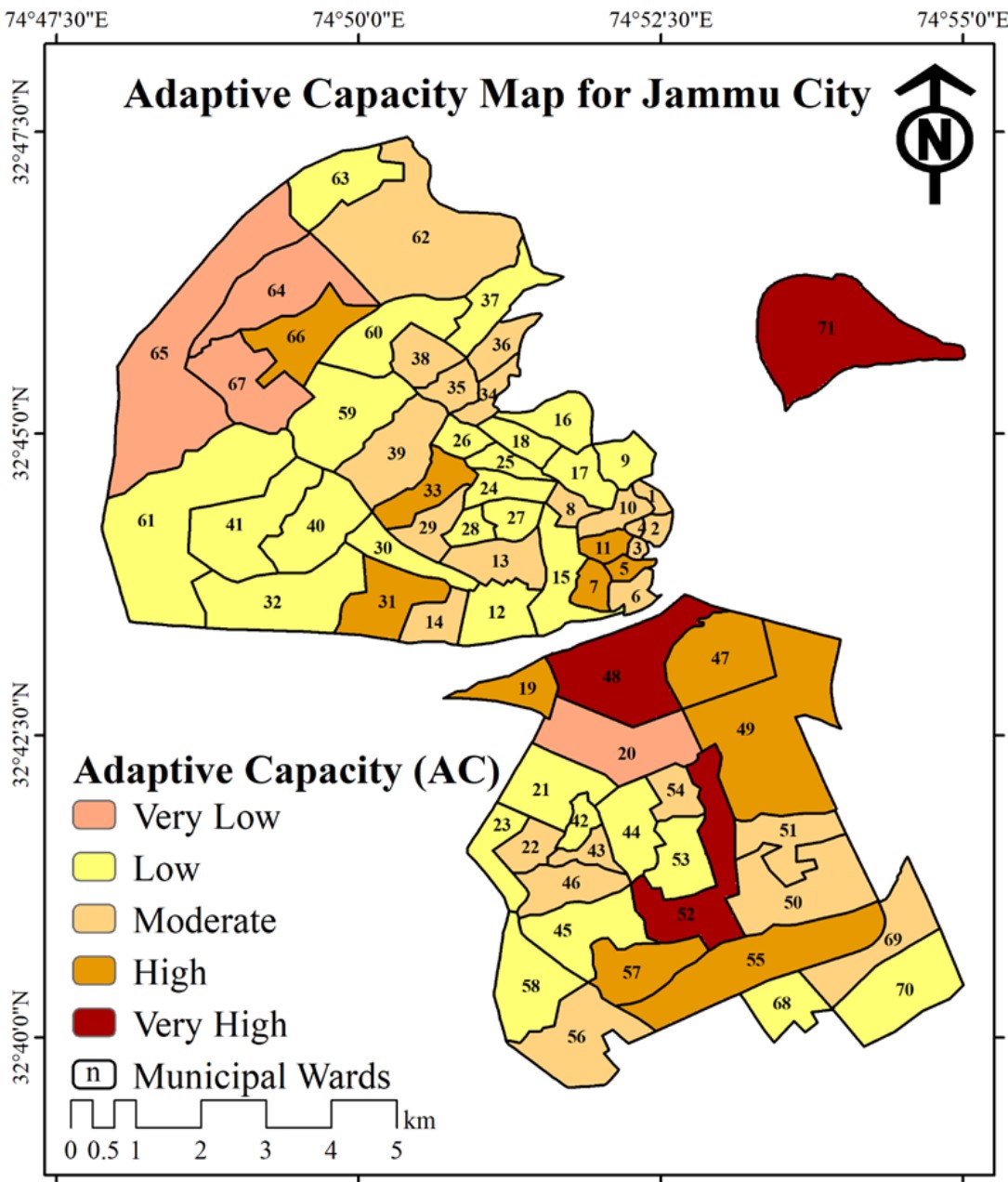

**Figure 13.** Spatial distribution of adaptive capacity map of Jammu City overlaid on municipal ward boundaries.

NDVI, i.e., the presence of vegetation, has been found to increase the adaptive capacity of a few wards in Jammu, particularly those in proximity to forests. Adapting cities to heat stress includes factors linked to vegetation, water, built form, and materials [70]. Keeping in view the adaptation responses, planners focus specifically on using vegetation to cool the urban environment [128]. It was found that areas with less availability of vegetation had a higher concentration of vulnerable population groups [129,130].

#### 4.4.4. Spatial Distribution of the Ward-Level Inherent Vulnerability Index (VI)

The values collected for the indicators were used to aggregate, integrate, and weigh the indices for the several primary components. The Vulnerability Index (VI) presented in Figure 14 shows the spatial distribution pattern of inherent vulnerability in the municipal wards of Jammu City. The Vulnerability Index (VI) was calculated using three variables: exposure, sensitivity, and adaptive capacity (Figures 11–13). The correlation analysis showed a positive relationship of VI with both exposure and sensitivity, having a value of 0.81. However, adaptive capacity showed a negative correlation with VI, with a value of −0.46. Among all the three variables, the high value of correlation of 0.81 indicates the role of exposure and sensitivity in influencing inherent vulnerability in Jammu City. The spatial distribution pattern of inherent vulnerability in the municipal wards of Jammu City is depicted by the Vulnerability Index (VI), which is shown in Figure 14. Similar findings have been noted by [131] in the Himalayan belt. Due to variations in the indicators' relative contributions to vulnerability, a thorough examination showed significant variations in the various components of vulnerability among the wards.

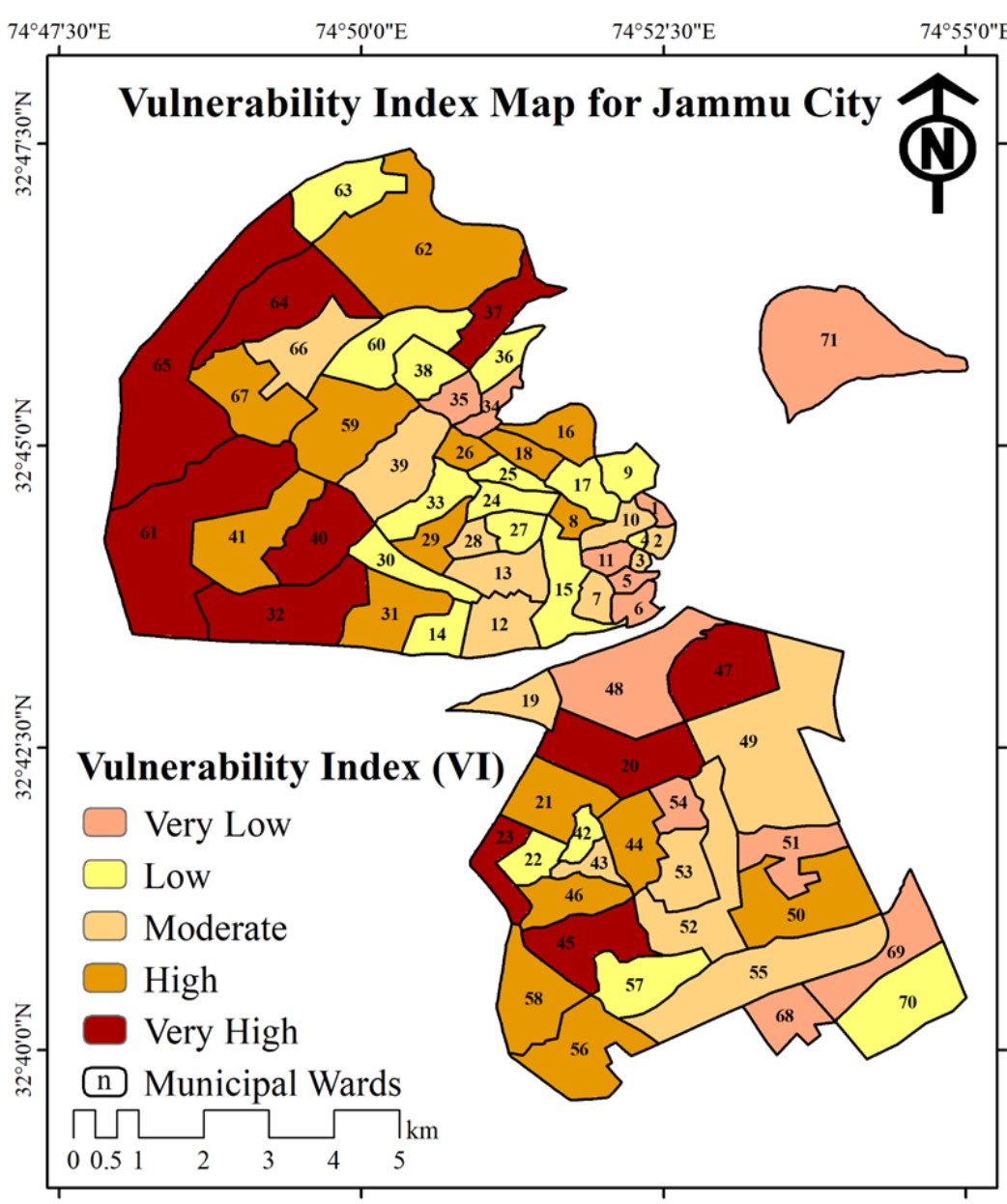

**Figure 14.** Spatial distribution of vulnerability index overlaid on municipal ward boundaries.

The vulnerability index is classified into five classes (very low, low, moderate, high, and very high), presented in Table 5, which shows that 36.53 sq.km (50.66%) of the total study area falls into the high to very high categories of vulnerability. The moderately vulnerable areas cover 20 percent of the study area, whereas low and very low constitute an area of 15.75% and 13.55%, respectively. The very highly vulnerable wards in the range of 0.48–0.59 are confined to northwestern areas of Jammu City, whereas few of them are in the core urban area. Out of a total of 10 wards that fall under the very highly vulnerable category, ward numbers W45 (Digiana), W65 (Barnai/Upper Dharmal), and W61 (Patta Paloura) displayed values on the higher side, i.e., 0.59, 0.57, and 0.54, respectively. In W45 and W61, the main drivers responsible for high vulnerability are population, non-workers, female population, electricity, pucca and semi-pucca structures, among all the three variables of exposure, sensitivity. and adaptive capacity. In W65. the drivers responsible for high vulnerability are home ownership, safe drinking water, pucca and semi-pucca structures, electricity, and built-up land, average size of household, and area of ward. The wards pertaining to low vulnerability are mainly due to low population density, no dependency on agriculture, good literacy rate, and health facilities, other than indicators which differ between wards, like low female and young population, non-workers, SC/ST population, proper sanitation, and safe drinking water. The household assets, pucca and semi-pucca structures, LST, and NDVI also contribute to low vulnerability in few of the wards.

Due to their large population, LST, and large ward area, wards 61, 65, 32, 23, 4, 2, 40, 57, 64, 47, and 56 experienced a significant impact from exposure. In contrast, wards 47, 45, 32, 61, and 40 had high sensitivity components, which had a cumulative effect on those wards and made them the most vulnerable (45, 65, 61, 32, 47, 40, 64, and 37, in order of degree of cumulative effect of vulnerability) (Figure 15). Nonetheless, a number of the municipal wards (48, 71, 52, 57, and 49) showed a high degree of adaptability and were hence less susceptible. The main indicators that went into creating these wards were the fair amount of greenery, the high literacy rate, the decent presence of health facilities, and the good possession of family assets.

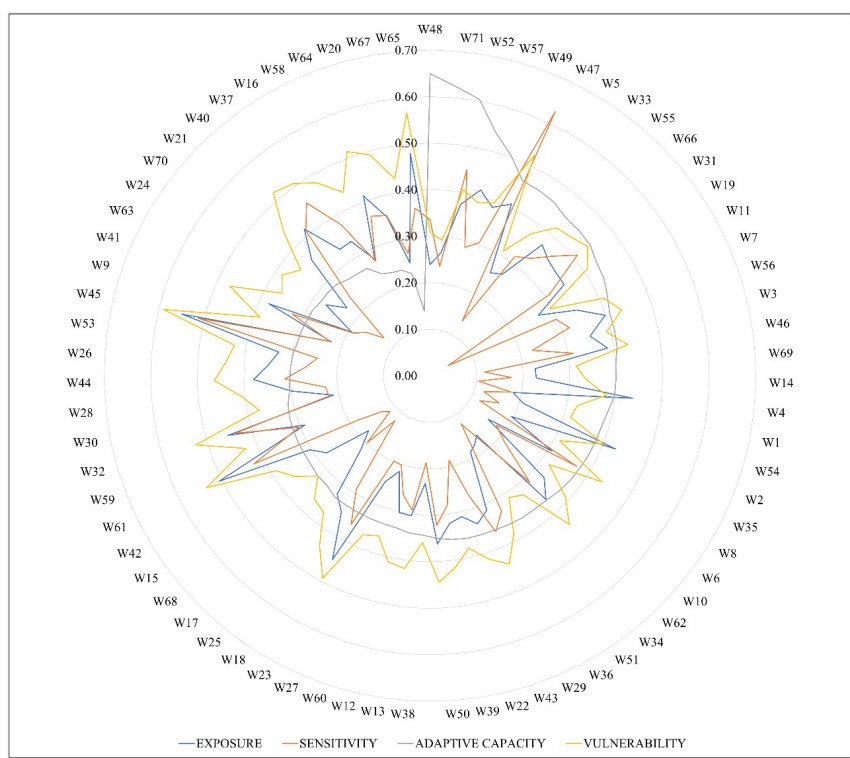

**Figure 15.** Relationship between Exposure, Sensitivity, Adaptive Capacity, and Vulnerability Index for Jammu City.

Current spatial planning practices—even general policies in various Indian cities—are predicated on erroneous and biased conclusions, as well as simplistic quantitative analyses that impede the ability of the public to make decisions and prevent equitable trade-offs between issues of priority and non-priority [132–134]. By determining the importance and contribution of each indicator by a thorough vulnerability assessment—which is also the main area of focus for spatial planning policies—this study attempts to address these problems. Furthermore, a comparative analysis can apply multiple modest interventions rather than a single dominant intervention, because the priorities are ranked both regionally and at the sector level.

This paper offers a logical and straightforward vulnerability assessment methodology that can be applied at any local spatial scale. It can be used to prioritize response measures on its own or as part of the spatial planning process. Nonetheless, each vulnerability component's indicator selection should be tailored to the needs of a particular area. This is an important phase because it has a large impact on the assessment's final result. The availability of crucial data and comprehensive information required by the vulnerability assessment framework posed a significant challenge for the use of the vulnerability assessment approach in the chosen case study.

## 5. Conclusions

The present study explains the spatial distribution of inherent vulnerability of Jammu City at ward level based on three important variables including exposure, sensitivity, and adaptive capacity. Due to intensified development in urbanized areas, diminished presence of water bodies and vegetation, along with other socioeconomic factors, Jammu City exhibits a high to very high vulnerability at the ward level; 10 wards within the municipal limits show very high vulnerability, while 16 wards were found to be in the high vulnerability index category. Alarmingly, around 50.66% of Jammu City overall has been classified under high to very high category of vulnerability. The spatial distribution of adaptive capacity, i.e., the ability to cope up in any kind of extreme event, was found to be very low, as 37.9% of the total area is in the low category. Although ward numbers W48, W52, and W71 displayed a high adaptive capacity value, the exposure level of W52 is high. Very high values of exposure were observed in W23, W32, W45, W61, and W65. The primary contributors to exposure are the growing population and elevated Land Surface Temperature (LST). Population expansion exerts heightened pressure on land resources, accelerates urbanization, diminishes vegetation coverage, and intensifies anthropogenic and industrial activities, consequently amplifying the generation of waste heat. In addition, apart from the higher vulnerability in Jammu City, it was also perceived that factors such as good literacy levels, the presence of pucca and semi-pucca structures, and good facilities such as household assets and health institutions can contribute increasing the adaptive capacity of the wards hence making them low vulnerable. This study serves as a baseline information to identify the area for focused adaptation strategies in order to cope with any extreme events. Future research can be carried out by integrating future vulnerability assessments, to offer more detailed insights.

**Author Contributions:** Conceptualization, S.B. and S.K.; methodology, M.F., F.M. and S.K; software, S.B., A.U.H. and S.K.S.; validation, S.B., A.U.H. and S.K.S.; formal analysis, S.B., A.U.H., F.M. and L.T.S.G.; investigation, S.B. and A.U.H.; resources, L.T.S.G. and P.K.; data curation, P.K.; writing—original draft preparation, S.B., S.K. and G.M.; visualization, L.T.S.G.; supervision and administration, M.F. and S.K.; writing—review and editing, S.B., A.U.H., F.M., M.F. and S.K.S. All authors have read and agreed to the published version of the manuscript.

**Funding:** This research received no external funding.

**Data Availability Statement:** The data pertaining to satellite image Landsat 8 OLI can be accessed from United States Geological Survey (USGS) at https://earthexplorer.usgs.gov/ (accessed on 14 January 2024). The data pertaining to socio-economic indicators is available at https://censusindia.gov.in (accessed on 14 January 2024), https://ecostatjk.nic.in/ (accessed on 14 January 2024) and https://jammu.nic.in/ (accessed on 14 January 2024).

**Conflicts of Interest:** The authors declare no conflicts of interest. All ideas expressed in this document do not necessarily represent the views of the organizations/institutions the authors belong to.

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
