# Peer review of "Development of Inherent Vulnerability Index within Jammu Municipal Limits, India"

_climate, doi:10.3390/cli12010012_

Round 1

Reviewer 1 Report

Comments and Suggestions for Authors

All the tables in the paper really need to be redone so that they are easier to read and more informative

Table 2 is especially awful. There is no information at all about each of the wards. This could be rectified by including that information in an appendix. However, the table itself is way too big and difficult for the reader. 

The findings need to be rewritten so that they provide more information and insight. Currently, they provide very little in meaningful dialogue about what was found in their research. There is some, but more insight is necessary

Author Response

We are grateful to respected reviewers for their valuable comments which helped to improvise this manuscript. As per the suggestions and comments of reviewers we have incorporated changes and modified the manuscript. The point wise comments and responses are given below:

#Reviewer 1

Comment1: All the tables in the paper really need to be redone so that they are easier to read and more informative.

Response: As per the suggestion of respected reviewers we have revised the tables so that it can look more informative.

Comment 2: Table 2 is especially awful. There is no information at all about each of the wards. This could be rectified by including that information in an appendix. However, the table itself is way too big and difficult for the reader.

Response: As per the suggestion of a respected reviewer we have modified all tables including table 2 so that it will be easy to read.

Comment 3: The findings need to be rewritten so that they provide more information and insight. Currently, they provide very little in meaningful dialogue about what was found in their research. There is some, but more insight is necessary.

Response: We have modified the results and discussion in line of suggestions given by the respected reviewer.

Reviewer 2 Report

Comments and Suggestions for Authors

The article "Development of Inherent Vulnerability Index with in Jammu Municipal Limits, India" is aimed at (1) deriving land use/land cover, land surface temperature, and normalized difference vegetation index using temporal datasets from Jammu city, India; (2) assessing the change in vulnerability of Jammu city using an indicator-based approach, and (3) assessing the inherent vulnerability of Jammu city at the ward level, considering exposure, sensitivity, and adaptive capacity. The three objectives answer three research questions: (1) how are and use/land cover, land surface temperature, and normalized difference vegetation index  interrelated across the temporal datasets, (2) what is the degree of change in vulnerability for Jammu city, and (3) what is the spatial variability of vulnerability among different wards of Jammu city? As it can be seen from the research goals and questions, the article presents a case study with no research depth. This is the main shortcoming of the manuscript. The article itself is interesting and could make an important contribution to the field, but in its current form the manuscript lacks research depth, visible by a focus on the case study rather than the research issue, its broader significance and contribution of findings to the theoretical advancement of the field, proved by poor introduction and discussions. In general, the manuscript presents too many results, without a clear indication of their contribution to the advancement of the field, and gives the impression of a case study report, and not of a scientific article. Most results are not exploited sufficiently, and make little sense for an international reader. Detailed comments are provided for each section of manuscript.
Normally, in a scientific article the introduction critically analyzes the existing literature in order to identify their shortcomings (ambiguities, controversies, misconceptions or lacks), justifying the need for research. However, in this manuscript the introduction lacks a critical touch; the fact that the authors derive land use/land cover, land surface temperature, and normalized difference vegetation index using temporal datasets from Jammu city fills in the lack of local information, but does not justify the publication in an international journal. In order to deserve being published in an international journal, an article should prove theoretical or methodological novelty valid beyond the case study, and being of interest to an international audience. Nevertheless, finding these novel elements require a better substantiation than the current one, based on only 45 references. The authors should expand their review, focusing on theoretical issues able to create a theoretical framework for their study.
Figure 1 shows the inability of authors to write up research. This is an article for an international journal, and not a report for the national authorities. The authors should present a map showing the location of the study area in an international context, making visible the neighboring countries with their names, so that a Brazilian researcher could understand it too.
The most important section of a research article, the Discussions, is insufficiently developed. The section is meant to emphasize the importance of research, justifying its publication. Normally, this section includes include (A) the significance of results - what do they say, in scientific terms; (B) the inner validation of results, against the study goals or hypotheses; (C) the external validation of results, against those of similar studies from other countries, identified in the literature; (D) the importance of results, meaning their contribution (conceptual or methodological) to the theoretical advancement of the field; (E) a summary of the study limitations and directions for overcoming them in the future research. Only the significance of results is presented, and to a very little extent. The "Discussions" should be developed to include the missing elements.
The conclusions are not entirely real conclusions, but only a summary of main findings. Conclusions are meant to deliver a scientific message, far away beyond the case study, to the entire scientific community, making a clear contribution to the theoretical (conceptual or methodological) development of the field. Conclusions must be cleansed of useless figures (lines 841-848).
The abstract looks like a shopping list, focusing on the case study only, and not on the broader implications of research and only on what has been done, without sufficient indications on why it has been done, and what knowledge gap is actually being filled in. The abstract is supposed to deliver ideas, and not state the research steps in brief, and provide useless figures (e.g., lines 25-29). It needs to be rewritten entirely, and shift the focus from a descriptive presentation to an analytical one.

Author Response

We are grateful to respected reviewers for their valuable comments which helped to improvise this manuscript. As per the suggestions and comments of reviewers we have incorporated changes and modified the manuscript. The point wise comments and responses are given below:

#Reviewer 2

Comment 1 : The article "Development of Inherent Vulnerability Index with in Jammu Municipal Limits, India" is aimed at (1) deriving land use/land cover, land surface temperature, and normalized difference vegetation index using temporal datasets from Jammu city, India; (2) assessing the change in vulnerability of Jammu city using an indicator-based approach, and (3) assessing the inherent vulnerability of Jammu city at the ward level, considering exposure, sensitivity, and adaptive capacity. The three objectives answer three research questions: (1) how are and use/land cover, land surface temperature, and normalized difference vegetation index interrelated across the temporal datasets, (2) what is the degree of change in vulnerability for Jammu city, and (3) what is the spatial variability of vulnerability among different wards of Jammu city? As it can be seen from the research goals and questions, the article presents a case study with no research depth. This is the main shortcoming of the manuscript. The article itself is interesting and could make an important contribution to the field, but in its current form the manuscript lacks research depth, visible by a focus on the case study rather than the research issue, its broader significance and the contribution of findings to the theoretical advancement of the field, proved by poor introduction and discussions. In general, the manuscript presents too many results, without a clear indication of their contribution to the advancement of the field, and gives the impression of a case study report, and not of a scientific article. Most results are not exploited sufficiently and make little sense for an international reader. Detailed comments are provided for each section of the manuscript.

Response: We have revised full manuscript in line of suggestions of respected reviewer

Comment 2:Normally, in a scientific article the introduction critically analyzes the existing literature to identify their shortcomings (ambiguities, controversies, misconceptions or lacks), justifying the need for research. However, in this manuscript the introduction lacks a critical touch; the fact that the authors derive land use/land cover, land surface temperature, and normalized difference vegetation index using temporal datasets from Jammu city fills in the lack of local information, but does not justify the publication in an international journal. In order to deserve being published in an international journal, an article should prove theoretical or methodological novelty valid beyond the case study, and being of interest to an international audience. Nevertheless, finding these novel elements require a better substantiation than the current one, based on only 45 references. The authors should expand their review, focusing on theoretical issues able to create a theoretical framework for their study. Figure 1 shows the inability of authors to write up research. This is an article for an international journal, and not a report for the national authorities. The authors should present a map showing the location of the study area in an international context, making visible the neighboring countries with their names, so that a Brazilian researcher could understand it too. The most important section of a research article, the Discussions, is insufficiently developed. The section is meant to emphasize the importance of research, justifying its publication. Normally, this section includes include (A) the significance of results - what do they say, in scientific terms; (B) the inner validation of results, against the study goals or hypotheses; (C) the external validation of results, against those of similar studies from other countries, identified in the literature; (D) the importance of results, meaning their contribution (conceptual or methodological) to the theoretical advancement of the field; (E) a summary of the study limitations and directions for overcoming them in the future research. Only the significance of results is presented and to a very little extent. The "Discussions" should be developed to include the missing elements. The conclusions are not entirely real conclusions, but only a summary of the main findings. Conclusions are meant to deliver a scientific message, far beyond the case study, to the entire scientific community, making a clear contribution to the theoretical (conceptual or methodological) development of the field. Conclusions must be cleansed of useless figures (lines 841-848).

Response: We have revised the full manuscript in line of suggestions of respected reviewer from abstract to conclusion along with figures and tables.

Comment 3:The abstract looks like a shopping list, focusing on the case study only, and not on the broader implications of research and only on what has been done, without sufficient indications on why it has been done, and what knowledge gap is actually being filled  in. The abstract is supposed to deliver ideas, and not state the research steps in brief, and provide useless figures (e.g., lines 25-29). It needs to be rewritten entirely, and shift the focus from a descriptive presentation to an analytical one.

Response: We have revised the abstract in modified manuscript in line of the suggestions given by respected reviewer

Reviewer 3 Report

Comments and Suggestions for Authors

1.      There is no clear statement how Authors understand vulnerability in theirs study? Is it social vulnerability or climate change vulnerability? Or combination of them? It is important for defining and justifying individual studied indicators.

2.      Defining of research gap is limited to Jammu city. Are there any examples of similar research regarding other cities in the world?

3.      Different population numbers are given in line 135 and 150. Different units are used in population description. What is the aim of presenting old data from 2011? (Figure 2). Moreover, in the section 2, analysed time period includes years 1992-2021. In Table 2 data for year 2021 are given.

4.      Section 2 is too long (12 pages)  and it should be divided into subsection, maybe according to the 4 phases of the study. In this way it would be more readable and clearly presented.  Moreover, transfer some data into annexes should be considered.

5.      Figure 3: What is justification for individual indicators’ selection? 

Author Response

We are grateful to respected reviewers for their valuable comments which helped to improvise this manuscript. As per the suggestions and comments of reviewers we have incorporated changes and modified the manuscript. The point wise comments and responses are given below:

#Reviewer 3

Comment 1: There is no clear statement how Authors understand vulnerability in theirs study. Is it social vulnerability or climate change vulnerability? Or a combination of them? It is important for defining and justifying individual studied indicators.

Response: We have defined the vulnerability in revised manuscript which is inherent vulnerability. In addition, we have incorporated the rationale for the selection of indicators.

Comment 2:Defining of research gap is limited to Jammu city. Are there any examples of similar research regarding other cities in the world? 3. Different population numbers are given in line 135 and 150.

Response: The literature review and research gap incorporated in revised manuscript.

Comment 3:Different population numbers are given in line 135 and 150. Different units are used in population description. What is the aim of presenting old data from 2011? (Figure 2). Moreover, in the section 2, analysed time period includes years 1992-2021. In Table 2 data for year 2021 are given.

Response: As pointed out by the worthy reviewer the numbers have been corrected and the necessary revisions have been done. The only population data available is from Census of 2011 as now new census has been carried out till date hence we were constrained to use the same.   

Comment 4:Section 2 is too long (12 pages) and it should be divided into subsection, maybe according to the 4 phases of the study. In this way it would be more readable and clearly presented. Moreover, transfer some data into annexes should be considered.

Response: we have revised the materials and methods section and divided the same into subsections.

Comment 5: Figure 3: What is justification for individual indicators’ selection?

Response: The rationale for selection of each indicator is given in revised manuscript

Round 2

Reviewer 2 Report

Comments and Suggestions for Authors

The authors have performed a very shallow revision of the manuscript, failing to address the critical points, able to justify the publication of their manuscript. This can be seen even inn the response, where they repeat the same general sentence written in an inappropriate boosting language. The manuscript lacks a critical review of the literature, and discussions revealing the significance of their results. Based on the revision, I believe that the authors lack the ability to address the comments and advocate the rejection of their manuscript.

Author Response

#Reviewer 2 (New Comment)

Comment: The authors have performed a very shallow revision of the manuscript, failing to address the critical points, able to justify the publication of their manuscript. This can be seen even in the response, where they repeat the same general sentence written in an inappropriate boosting language. The manuscript lacks a critical review of the literature, and discussions revealing the significance of their results. Based on the revision, I believe that the authors lack the ability to address the comments and advocate the rejection of their manuscript.

Response: As confirmed by reviewer 3 entire manuscript has been extensively revised which is evident upon comparing the previously submitted manuscript. However, the revision is entirely based on the suggestions and comments provided by the reviews.

#Reviewer 2 (Original Comments)

Comment 1 : The article "Development of Inherent Vulnerability Index with in Jammu Municipal Limits, India" is aimed at (1) deriving land use/land cover, land surface temperature, and normalized difference vegetation index using temporal datasets from Jammu city, India; (2) assessing the change in vulnerability of Jammu city using an indicator-based approach, and (3) assessing the inherent vulnerability of Jammu city at the ward level, considering exposure, sensitivity, and adaptive capacity. The three objectives answer three research questions: (1) how are and use/land cover, land surface temperature, and normalized difference vegetation index interrelated across the temporal datasets, (2) what is the degree of change in vulnerability for Jammu city, and (3) what is the spatial variability of vulnerability among different wards of Jammu city? As it can be seen from the research goals and questions, the article presents a case study with no research depth. This is the main shortcoming of the manuscript. The article itself is interesting and could make an important contribution to the field, but in its current form the manuscript lacks research depth, visible by a focus on the case study rather than the research issue, its broader significance and the contribution of findings to the theoretical advancement of the field, proved by poor introduction and discussions. In general, the manuscript presents too many results, without a clear indication of their contribution to the advancement of the field, and gives the impression of a case study report, and not of a scientific article. Most results are not exploited sufficiently and make little sense for an international reader. Detailed comments are provided for each section of the manuscript.

Response: Since the study is the outcome of a policy document, though a case study, our study presents a very important aspect that is lacking in the present-day planning process. The climate resilient action plan of Jammu City is based on the important vulnerability assessment outcomes of this study which shall not only prove beneficial for other cities but also an important tool for the adaptation planning process for government planners. As far as the research depth of the study is concerned, Himalayan regions are considered blackspots of data, and the selection of indicators and designing research experiments on these indicators is extremely challenging, we believe that we have followed the research methodology framework prescribed by IPCC and the paper has significance at City level assessments.

We have improved the abstract, introduction, and methodology section significantly with an extensive review of the literature. The results have been made crisp and clear now as per the suggestions. The results also bring out the intervention areas for planners to improve the adaptive capacity of individual wards for focused planning. Hence contributing greatly to research areas for linking science with policy.

Comment 2: Normally, in a scientific article the introduction critically analyzes the existing literature to identify their shortcomings (ambiguities, controversies, misconceptions or lacks), justifying the need for research. However, in this manuscript the introduction lacks a critical touch; the fact that the authors derive land use/land cover, land surface temperature, and normalized difference vegetation index using temporal datasets from Jammu city fills in the lack of local information, but does not justify the publication in an international journal. In order to deserve being published in an international journal, an article should prove theoretical or methodological novelty valid beyond the case study, and being of interest to an international audience. Nevertheless, finding these novel elements require a better substantiation than the current one, based on only 45 references. The authors should expand their review, focusing on theoretical issues able to create a theoretical framework for their study. Figure 1 shows the inability of authors to write up research. This is an article for an international journal, and not a report for the national authorities. The authors should present a map showing the location of the study area in an international context, making visible the neighboring countries with their names, so that a Brazilian researcher could understand it too. The most important section of a research article, the Discussions, is insufficiently developed. The section is meant to emphasize the importance of research, justifying its publication. Normally, this section includes include (A) the significance of results - what do they say, in scientific terms; (B) the inner validation of results, against the study goals or hypotheses; (C) the external validation of results, against those of similar studies from other countries, identified in the literature; (D) the importance of results, meaning their contribution (conceptual or methodological) to the theoretical advancement of the field; (E) a summary of the study limitations and directions for overcoming them in the future research. Only the significance of results is presented and to a very little extent. The "Discussions" should be developed to include the missing elements. The conclusions are not entirely real conclusions, but only a summary of the main findings. Conclusions are meant to deliver a scientific message, far beyond the case study, to the entire scientific community, making a clear contribution to the theoretical (conceptual or methodological) development of the field. Conclusions must be cleansed of useless figures (lines 841-848).

Response: In light of comments a section-wise detailed response is given below:

  1. Entire manuscript has been subjected to an extensive revision with modifications in abstract, introduction and lucidity in methodology.
  2. The review of literature has been revised after consulting enormous number of research articles i.e., 135 references. (Line 46-99)
  3. We have addressed the write-up issues to differentiate between report and research article. However, we believe that the results are bringing out the intervention areas for planners for improving the adaptive capacity of individual wards for focused planning. Hence contributing greatly in research area for linking science with policy. The researchers have been always craving for use of their work in policy planning, we feel greatly honored that this science-based input has been put in use by the local planners.
  4. The study area map has been revised as per the suggestion of the reviewer. (Line 134-135).
  5. The discussion complementing results and their summary has been revised as per the comments of the reviewer. The discussion has been substantiated with the findings of other researchers with appropriate references. (Lines 232-554).
  6. The conclusion has been fully revised with contributions in the scientific as well as policy making arena besides recommendations have also been added as per the suggestions of the reviewer. (Line 556-577)

Comment 3: The abstract looks like a shopping list, focusing on the case study only, and not on the broader implications of research and only on what has been done, without sufficient indications on why it has been done, and what knowledge gap is actually being filled in. The abstract is supposed to deliver ideas, and not state the research steps in brief, and provide useless figures (e.g., lines 25-29). It needs to be rewritten entirely, and shift the focus from a descriptive presentation to an analytical one.

Response: We have revised the abstract in modified manuscript focusing on all the aspects of the paper as per suggestions given by reviewer. (Line 22-41 entirely revised)

Reviewer 3 Report

Comments and Suggestions for Authors

My all remarks for the first version of the manuscript were considered by the Authors. The article was rewritten in it's significant part. 

Please look at page numbering, because it is mistake starting from page 10.

Author Response

Comments and Suggestions for Authors

My all remarks for the first version of the manuscript were considered by the Authors. The article was rewritten in its significant part. 

Please look at page numbering, because it is a mistake starting from page 10.

Response: Thanks for your positive comments. We have revised all page numbers and Line numbers as suggested.